# Microbial Spoilage of Traditional Goose Sausages Produced in a Northern Region of Italy

**DOI:** 10.3390/microorganisms11081942

**Published:** 2023-07-29

**Authors:** Michela Pellegrini, Federica Barbieri, Chiara Montanari, Lucilla Iacumin, Cristian Bernardi, Fausto Gardini, Giuseppe Comi

**Affiliations:** 1Department of Agricultural, Food, Environmental and Animal Science, University of Udine, 33100 Udine, Italy; pellegrini.michela@spes.uniud.it (M.P.); lucilla.iacumin@uniud.it (L.I.); 2Department of Agricultural and Food Sciences, University of Bologna, 47521 Cesena, Italy; federica.barbieri16@unibo.it (F.B.); chiara.montanari8@unibo.it (C.M.); fausto.gardini@unibo.it (F.G.); 3Department of Veterinary Medicine and Animal Sciences, University of Milan, 20122 Lodi, Italy; cristian.bernardi@unimi.it

**Keywords:** *Levilactobacillus brevis*, mould, enterococci, spoilage, goose sausage

## Abstract

Recently, during the ripening of goose sausage, a defect consisting of ammonia and vinegar smell was noticed. The producer of the craft facility, located in Lombardia, a Northern region of Italy, asked us to identify the cause of that defect. Therefore, this study aimed to identify the potential responsible agents for the spoilage of this lot of goose sausages. Spoilage was first detected by sensory analysis using the “needle probing” technique; however, the spoiled sausages were not marketable due to the high ammonia and vinegar smell. The added starter culture did not limit or inhibit the spoilage microorganisms, which were represented by *Levilactobacillus brevis*, the predominant species, and by *Enterococcus faecalis* and *E. faecium*. These microorganisms grew during ripening and produced a large amount of biogenic amines, which could represent a risk for consumers. Furthermore, *Lev. brevis*, being a heterofermentative lactic acid bacteria (LAB), also produced ethanol, acetic acid, and a variation in the sausage colour. The production of biogenic amines was confirmed in vitro. Furthermore, as observed in a previous study, the second cause of spoilage can be attributed to moulds which grew during ripening; both the isolated strains, *Penicillium nalgiovense*, added as a starter culture, and *P. lanosocoeruleum*, present as an environmental contaminant, grew between the meat and casing, producing a large amount of total volatile nitrogen, responsible for the ammonia smell perceived in the ripening area and in the sausages. This is the first description of *Levilactobacillus brevis* predominance in spoiled goose sausage.

## 1. Introduction

In Italy, pork, beef, wild game (deer), and poultry meats are often used to produce a large number of traditional sausages. In Northern regions, from Lombardia to Friuli Venezia Giulia, goose meat represents one of the main ingredients for sausages, characterised by a sour taste and a semirigid consistency, which is elastic but not rubbery [1].

Goose products have always been present in the gastronomic tradition of different Northern Italy areas. The presence of Jewish communities in these areas has been documented since the fifteenth century, and the most accredited hypothesis for the widespread presence of this type of sausage in these areas is the presence since the fifteenth century of Jewish communities whose religion forbids the consumption of pork meat; therefore, they had to consume meat coming from other animals like goose, whose breeding is favoured by the characteristics of the territory.

Traditionally, the ingredients include fresh or frozen goose meat, NaCl, additives (nitrates, nitrites), spices, dextrose, and microbial starters, mainly represented by coagulase-negative, catalase-positive cocci (CNCPC) and *Latilactobacillus sakei*. However, the recipe varies according to the producers. The quality of these sausages is not sufficiently standardised because they are mainly produced by shops, farms, typical restaurants, or by artisanal facilities, which are not equipped with appropriate drying and ripening chambers or systems with complete control of relative humidity (RH) and temperature. Therefore, each lot of production has its own history and can be completely different from other lots. However, the choice of raw material, the natural microclimate of the drying/ripening rooms, and the aptitude of the producers allow to obtain a high-quality product [1]. The goose sausage ripening depends on microbial and tissue enzyme activities and it is similar to that of sausages made with pork meat and fat [1,2,3,4].

Among the microbial population, CNCPC and homofermentative lactic acid bacteria (LAB) are the main microorganisms responsible for ripening [1,5,6,7,8,9,10,11]. These bacteria originate from raw meat, but they can be intentionally added as microbial starters to ensure consistent aroma and flavour, improve quality and safety, reduce the length of the curing period, and standardise production [1,2,3,7,8,12,13,14]. Goose sausages are ancient and traditional products, and recently, their demand has increased because consumers are searching for new foods; they represent an effort to generate alternatives to beef and pork meat products. While chicken sausages are often mixed with meat obtained from other animals, the combination of goose meat with other meat is quite rare; generally, goose sausages are produced using only goose meat [15]. In Italy, recently, the population has rediscovered products based on regional recipes, and goose sausages constitute an example [1]. The quality of craft sausages can depend on various parameters, including goose breeding, slaughtering, and sausage manufacturing. However, goose breeding and slaughtering are easily checked depending on strict and established procedures. Indeed, in craft facilities, the main parameters influencing the quality are particularly caused by the production processes, which are often insufficiently checked, making the sausages unsuitable for consumption. The failure in the production process depends on the applied technology (temperature/relative humidity of the ripening rooms), the natural microbiota of the meat, and the ingredients. The defects, coming from inadequate ripening, often lead to unpleasant odours or tastes. Ammonia, sulfur compounds, ketones, aldehydes, esters, organic acids, and biogenic amines represent the main molecules produced by spoilage microorganisms in sausages [1,16,17,18,19]. In particular, ammonia compounds lead to an increase in pH and to off-odours and off-flavours [1,17]. Enterobacteria, moulds, and LAB are often recognised as responsible for producing ammonia and acetic acid, both indicators of sausage spoilage. Lactic acid bacteria can produce ammonia compounds only when a low level of sugar is added or present in the meat batter [16]; conversely, LAB produce lactic acid [1,16]. The presence of acetic acid, which gives a sour smell to sausages, is produced by enterobacteria, moulds, and heterofermentative LAB [1].

In fermented sausages, enterococci, *Enterobacteriaceae*, and some LAB are usually the main producers of biogenic amines (BAs), such as tyramine, histamine, cadaverine, and putrescine [1,19,20,21], which, being vasoactive, at ingestion levels of 50–100 mg [22] can cause hypertension, migraine, brain haemorrhages, heart failure [23], urticaria, headache, flushing, and abdominal cramps [20,24] in susceptible consumers.

Traditional microbiological methods, based on phenotypical aspects, are not useful for the early detection of BA producers in fermented sausages because only some strains of certain species can decarboxylate amino acids, such as tyrosine or histidine [20,25]. Indeed, only by molecular methods, it is possible to amplify all genes responsible for BA production [20,25,26,27]. Furthermore, in fermented sausages, BA formation depends on many factors, including salt concentration, pH, quality of the raw materials, starter culture, technological additives, packaging, irradiation, high-pressure treatment, diameter of the sausages, high initial concentration of decarboxylating microorganisms, inadequate technological processes, and the possible presence of antimicrobial compounds in the spices mixture used in sausage production [1,19,25,28,29,30,31,32,33,34,35].

Recently, during the ripening of goose sausage, a defect consisting of ammonia and vinegar smell was noticed and subsequently confirmed by a sensory analysis made by untrained panellists. The producer of the craft facility asked us to identify the cause of the defect. Therefore, this study aimed to explore the potential responsible agents for the spoilage of this lot of goose sausage.

## 2. Materials and Methods

### 2.1. Goose Sausage Products Analysed

Two lots of sausages (50 each) with the same goose meat and recipe were produced by a small-scale facility located in Lombardy. These lots differed for the added starter culture mix (A and B), both composed of *Staphylococcus xylosus* and *Latilactobacillus sakei* (1/1 ratio) but of two different lots of production. The sausages had the following composition: goose meat, 97%; NaCl, 2.8%; KNO_3_, 0.02%; dextrose, 0.2%; black pepper, 0.002%; and nutmeg, 0.002%. Before adding the ingredient of the recipe, one lot was added with a starter from lot A and one with a starter from lot B, both at a final concentration of 6 log CFU/g. A starter culture of *Penicillium nalgiovense* was spread by aerosol (approximately 3 log CFU/cm^2^) onto casings. Natural casings derived from the small intestine deprived of the mucosa were used.

The production process consisted of a dehydration phase for 7 days with a relative humidity (RH) between 65% and 85% at 18–20 °C and a ripening phase for 40 days with an RH of approximately 70% at 15–17 °C. Ten samples of both spoiled and unspoiled sausages were collected and used for further analyses.

Before the analysis, samples were washed to eliminate moulds on the casing, which was then aseptically removed, and then each sausage was sterile sliced.

During the ripening, the head of the sausage production checked both lots, but 10 days before the end of ripening, which lasted 50 days, the sausages with added lot B starter gave off an ammonia and vinegar smell, which was also widespread in the ripening rooms. Consequently, he identified this lot B as spoiled and asks us to identify the causes. The defect was confirmed by the sensory analysis listed below.

Conversely, the sausages supplemented with lot A starter gave off a typical smell of goose sausages and were identified as unspoiled. Therefore, considering the added starters, the sausages were named lot A and lot B.

### 2.2. Sensory Analysis

Both lots of sausages, those presumed spoiled and unspoiled, were evaluated by the “needle probing” technique by 15 panellists of a nonprofessional panel (all workers of the facility). The technique involves the rapid insertion of a thin horse bone into the sausages, resulting in the perception of odours [36]. Then the presumed spoiled and unspoiled sausages were sliced and subject to sensory analysis using the triangle test methodology (ISO 4120:2004, Triangle test methodology. Standard test method for sensory analysis—General guidance for the design of test rooms). Briefly, the untrained panellists were presented with three products, two of which were identical. Then they were asked to identify which product they believed to be a unique sample. The panellists who detected the presence of two distinct samples by smell were asked to identify which one was the best or the acceptable sample, considering only the odour parameter. 

### 2.3. Total Volatile Basic Nitrogen (TVB-N), pH, Acetic Acid, and Colour Determination

TVB-N was evaluated by the Pearson [37] method. Briefly, TVB-N was released by boiling the sample directly with magnesium oxide, which also prevented volatile acids from distilling over into boric acid. The distillate was titrated with standard acid. The pH of the product was measured directly by inserting a pH meter probe (Radiometer, København, Denmark) into the sample. The water activity (Aw) was determined using a Hygromer AWVC (Rotronic, Milan, Italy). Lactic acid was detected using a lactic acid kit (R-Biopharm, Milan, Italy) according to the manufacturer’s instructions. The final values of all the above physicochemical parameters were expressed as the respective average of measurements of six samples.

The colour was measured using a Minolta Chromameter CR-200 (Singapore) and the CIE Lab system. After calibration with standard white tiles, the chromameter was positioned perpendicular to the patty surface, and 10 different positions were evaluated for each of the 4 samples obtained immediately after slicing. The evaluated parameters were lightness (L*), red/green chromaticity (a*), and yellow/blue chromaticity (b*). Indeed L* describes the white intensity or brightness, with values ranging from 0 (black) to 100 (white); a* describes the redness (a* > 0); and b* describes the yellowness (b* > 0). In addition, the ΔE parameter between the data of spoiled and unspoiled products was also evaluated [38]. To simplify the results, the data based on the parameter ΔE were also reported, which allows for evaluating the difference between two colours. Here, the ΔE value at and between 0 and 60 days of storage of spoiled and unspoiled goose sausages was compared. The standard perception ranges are as follows: ≤1.0, not perceptible by the human eye; 1–2, perceptible through close observation; 2–10, perceptible at a glance; 11–49, colours are more similar than the opposite; and 100, colours are exactly the opposite.

### 2.4. Microbiological Analysis

Ten grams of each goose sausage was transferred to a stomacher bag, and 90 mL of saline sterile solution (0.9% *w*/*v*) was added. The sample was homogenised in a stomacher for 2 min. Appropriate decimal dilution was prepared and plated onto selective media to count different groups of microorganisms: The total viable count (TVC) was evaluated on plate count agar (PCA; Oxoid, Milan, Italy) incubated at 30 °C for 48–72 h; LAB were grown on De Man Rogosa Sharpe agar (MRS;, Oxoid, Milan, Italy) incubated at 42 °C for 48 h in anaerobic conditions; yeasts and moulds were grown on malt agar (MA; Oxoid) incubated at 25 °C for 72–96 h [39]; Escherichia coli was grown on violet red bile lactose agar (VRBLA; Oxoid) incubated at 44 °C for 24 h; Enterobacteriaceae were grown on violet red bile glucose agar (VRBGA; Oxoid) incubated at 37 °C for 24 h; coagulase-positive, catalase-positive cocci (CPCPC) were grown on Baird-Parker agar medium (BP; Oxoid) supplemented with egg yolk tellurite emulsion (Oxoid) incubated at 35 °C for 24–48 h and confirmed by a coagulase test; coagulase-negative, catalase-positive cocci (CPCNC) were grown on mannitol salt agar (MSA; Oxoid) incubated at 30 °C for 48 h; enterococci were grown on kanamycin aesculin azide agar (KAA; Oxoid) incubated at 37 °C for 48 h; sulfite-reducing clostridia were quantified on differential reinforced clostridia medium (DRCM; VWR, USA) incubated at 37 °C for 24–48 h in anaerobic conditions obtained with an anaerobic kit (BD BBL 261205 CO_2_ generators–GasPack jars–Becton Dickinson, Franklin Lakes, NJ, USA). *Salmonella* spp. was evaluated by the ISO (6579-1 2002 Cor.1:2004 Microbiology of food and animal feeding stuff—Horizontal method for the detection of *Salmonella* spp.) method and Listeria monocytogenes by another ISO (11290-1,2:1996 Adm.1:2004. Microbiology of food and animal feeding stuff—Horizontal method for the detection of Listeria monocytogenes).

### 2.5. Isolation and Identification of LAB

Thirty-nine colonies were randomly isolated from KAA agar plates and 258 from MRS agar plates of the spoiled and unspoiled sausages (total 297 isolates) containing 10 to 50 colonies and streaked onto the same new media.

After purification, the colonies were subjected to Gram staining and to a catalase test and then stored at −20 °C in MRS broth containing 30% (*v*/*v*) glycerol until molecular identification and characterisation. Gram-positive streptococci and catalase-negative colonies were identified by API 20 Strep according to the manufacturer’s method (BioMerieux Marcy-l’Étoile, France). Gram-positive rod, catalase-negative colonies (LAB) were identified by the method reported in [40].

### 2.6. Isolation and Identification of Moulds

Fifty-two mould colonies grown on MA were isolated from the spoiled goose sausages, purified and transferred onto three different agar media: Czapek Dox Agar (Oxoid, Milan, Italy), MA, and salt-malt agar (5% malt extract, 5% NaCl, distilled water 1 mL, pH 6.2; Oxoid, Milan, Italy). According to Samson et al. [39], the moulds were identified by morphological characters by macroscopic and microscopic examination (hyphae, spores and reproduction, colour of colony, and type of mycelium). The identification was confirmed by PCR-DGGE and sequencing according to the method reported by Iacumin et al. [41]. Briefly, the DNA of each colony was amplified by nested PCR (2-step amplification). Each amplicon was run in an acrylamide gel (DGGE), excised by gel cutting tips, and subjected to reamplification with the same primers without a GC clamp. The product was cloned into the pGEM-T easy vector (Promega, Milan, Italy) following the instructions of the manufacturer. The insert of the appropriate clone was sequenced by a commercial facility (Eurofins MWG GmbH, Martinsried, Germany). Sequence comparisons were performed using the Blast program [42].

### 2.7. In Vitro Reproduction of the Defect by Moulds

One hundred grams of the unspoiled meat homogenates of goose sausages were boiled in water (200 mL) for 1 h. After boiling, the mixture was filtered through cotton wool and sterilised at 115 °C for 15 min. The sterilised mixture was adjusted to 300 mL with distilled sterile water and distributed among 10 Petri plates (30 mL each). A loop of each isolated mould species was inoculated in the plates (one strain per plate), which were incubated for 7 days at 25 °C. Three replicates of each strain were performed. At the end of the incubation period, each mixture was filtered and analysed for the presence of TVB-N (total volatile basic nitrogen), biogenic amines, and acetic acid [1].

### 2.8. Volatile Compound Determination

The sausage volatile organic compound (VOC) profile was assessed using gas chromatograph–mass spectrometry coupled with solid-phase microextraction (GC-MS-SPME). A known amount of 4-methyl-2-pentanol (Sigma-Aldrich, Steinheim, Germany) as an internal standard was added to 3 g of sample and analysed according to the protocol reported by Bancalari et al. [43]. The identification of the VOC peaks was achieved using the Agilent Hewlett–Packard NIST 2011 mass spectral library (Gaithersburg, MD, USA) [44], and the data are expressed as the ratio between each molecule’s peak area and the peak area of the internal standard. The results are the mean of six determinations for each sample.

### 2.9. Biogenic Amines In Vitro and in Spoiled and Unspoiled Sausages

Twenty colonies of each isolated species were tested for biogenic amine production on agar media, according to the Bover-Cid and Holzapfel [45] method. Three spoiled and unspoiled meat homogenates were randomly sampled to detect the biogenic amines using the following method: The samples were extracted with trichloroacetic acid (5%), following the method described by Pasini et al. [46]. After dansyl-chloride derivatisation (Sigma Aldrich, Gallarate, Italy) [47], the extracts were analysed in an Agilent Technologies 1260 Infinity HPLC equipped with an automatic injector (G1329B ALS 1260, loop of 20 µL) and with a UV detector (G1314F VWD 1260) set at 254 nm, according to the method reported by Barbieri et al. [48]. The amounts of amines were expressed as mg/kg with reference to a calibration curve obtained through aqueous dansyl-chloride-derivatised amine standards of concentrations ranging from 10 to 200 mg/L (Sigma-Aldrich, Milano, Italy). The detection limit for all the amines was 3 mg/kg sample under the adopted conditions.

### 2.10. Statistical Analysis

Data were analysed using Statistica 7.0 version 8 software (Statsoft Inc., Tulsa, OK, USA, 2008). The values of the different parameters were compared by one-way analysis of variance, and the means were then compared using Tukey’s honest significance test. Differences were considered significant at *p* < 0.05. Each physical–chemical and microbial analysis included 10 samples either for spoiled or unspoiled goose sausages. Three samples were tested for volatilome analysis. 

## 3. Results

The ammonia smell perceived in the lot B ripening area was also confirmed in the product by all the nonprofessional panellists (15 workers of the facility) by the “needle probing” technique and by careful sniffing. All the panellists confirmed the presence of off-odours and off-flavours, particularly a vinegar odour, and marked the product as spoiled. Consequently, lot B was investigated to determine the cause of the presence of off-odours and off-flavours, and the data were compared with those of lot A sausages, whose smell was typical of unspoiled goose sausages.

Table 1 and Table 2 present the microbial and physicochemical characteristics of the unspoiled and spoiled goose sausages. The total microbial counts (TMC) and LAB counts were typical of ripened sausages. There was a significant difference (*p* < 0.05) in the TMC between the unspoiled and spoiled sausages, while no significant difference in the LAB count was observed. In particular, the means of TMC of both sausages were approximately 7.8 and 8.6 log CFU/g in unspoiled and spoiled sausages, respectively, and the means of the LAB counts were approximately 8.6 and 8.7 log CFU/g, respectively. Yeast and CNCPC concentrations also did not differ significantly (*p* > 0.05) between the spoiled and unspoiled sausages, considering the standard deviations. The yeast counts were at levels of 2.0 and 2.2 log CFU/g, and the CNCPC concentration was approximately 6 log CFU/g; both loads were also typical of traditional Italian sausages. The mould and enterococci concentrations differed significantly between the spoiled and unspoiled samples (*p* < 0.05), while *Enterobacteriaceae* did not (*p* > 0.05). The values of enterococci were less than 100 UFC/g in the unspoiled samples and 7.5 log CFU/g in the spoiled samples. The moulds in the unspoiled sausages were one log CFU/g lower than those in the spoiled sausages. However, the higher level of moulds in the spoiled sausages was due to their growth in the space between the meat and casing and did not depend on contamination during sampling because the casings of both groups were first brushed and washed. Indeed, in the spoiled sausages, the dehydration left spaces between the meat and casing, and just in these spaces mould could grow. Consequently, a white mould mycelium could be observed and sampled during inspection only in the spoiled sausages. This presence proved and demonstrated the higher level of moulds in spoiled sausages. *Enterobacteriaceae* reached 2.1 log CFU/g and 2.0 log CFU/g in the unspoiled and spoiled samples, respectively. CPCPC, sulfite-reducing Clostridia, and *E. coli* were present at concentrations below the detection limit of the method (LOD < 10 CFU/g). *Listeria monocytogenes* was present at less than 100 CFU/g, and *Salmonella* was absent in a 25 g sample, according to REG. EC 2073/05 (15/11/2005, L 338/1).

Moreover, the TVB-N value of the spoiled sausages reached 302.5 mg N/100 g and was approximately three times more than that of the unspoiled sausages (78.2 mg N/100 g).

The pH differed and was about 5.9 in unspoiled sausages and 6.3 in spoiled sausages (*p* < 0.05); in contrast, the Aw was similar at a level of 0.92, depending on the ripening methods, which was similar for the spoiled and unspoiled sausages.

The different pH values between the two types of sausages were confirmed by the analysis of the lactic acid concentration. In the unspoiled sausages, the mean of lactic acid concentration was 120.4 mg/100 g, which was higher than that of the unspoiled sausages (Table 2), and it remained at 85.3 mg/100 g, showing a significant difference (*p* < 0.05).

The colour of the spoiled sausages, as expressed by the evaluation of L*, a*, and b*, was not significantly different (*p* > 0.05) from that of the unspoiled sausages (Table 2). However, also the ΔE parameter was not different between the spoiled and unspoiled products. The ΔE value for both the products was at a level of 2.5; consequently, the colour of the two types of sausages was perceptible at a glance. However, the colour of the spoiled sausage appears clearer than that of the unspoiled sausage, and this depends on natural oxidative phenomena induced by heterofermentative LAB, which predominates in these types of sausages and is involved in spoilage.

The concentration of BA detected in the two samples is reported in Table 3. Spermine, spermidine, and 2-phenylethylamine contents were always below the limit of detection (<3 mg/kg). Histamine was identified only in the unspoiled sausages but at a low level (5.6 mg/kg). The spoiled products had high concentrations of putrescine (242.5 mg/kg), cadaverine (510.4 mg/kg), and tyramine (388.3 mg/kg), while in the unspoiled products, only low concentrations of cadaverine (8.1 mg/kg) were detected. Despite the absence of histamine in the spoiled sausages, which is the only amine subjected to the limitation in fish and fishery products (REG. EEC 2073/05), the relevant presence of other amines (especially tyramine) can be considered worrying in relation to EFSA indications [49].

It is well known that the release of small amounts of hydrogen peroxide and hydrogen sulfide by heterofermenting LAB produces discolouration and sometimes greening [1,50]. Only an evident discolouration was perceptible at a glance in the spoiled samples.

Table 4 presents the microbial species found in unspoiled or spoiled sausages.

In the unspoiled sausages, the isolates were identified as *Latilactobacillus sakei* (130 isolates), which predominated in all the sampled dilutions. Additionally, *Latilactobacillus curvatus* (4 isolates) was identified but only at 10^−6^ serial decimal dilution.

In spoiled sausages, the isolates were identified as *Levilactobacillus brevis* (97 isolates), which is the most present at decimal dilutions from 10^−6^ to 10^−8^. Enterococci were also isolated, particularly *Enterococcus faecium* (26 isolates) and *E. faecalis* (13 isolates), which were both detected from 10^−6^ to 10^−7^ decimal dilution. The predominance of *Latilactobacillus sakei* in the unspoiled sausages was expected because it was contained in the added starter; consequently, its presence demonstrated that the starter culture has worked. Conversely, in the spoiled sausages, the predominance of *Levilactobacillus brevis* followed by *E. faecium* and *E. faecalis* indicated that the starter did not work. Indeed, in these sausages, *Latilactobacillus sakei* (25 isolates) was detected only at decimal dilution from 10^−6^ to 10^−7^.

Table 5 presents the ability to produce biogenic amines in vitro of the randomly selected isolates (20 isolates for each species) issued from spoiled sausages. *E. faecium, E. faecalis*, and *Lev. brevis* are typical sugar-fermenting LAB, but they are also able to decarboxylate amino acids and produce amines. As observed in vitro, all the selected strains of *E. faecium* and *E. faecalis* produced tyrosine, cadaverine, and putrescine, while the *Lev. brevis* strains produced tyrosine and cadaverine. These data justify the presence of these biogenic amines found in the spoiled sausages. All the selected strains of *Lat. curvatus* produced only tyrosine, while the selected strains of *Lat. sakei* did not produce any amines in vitro.

The isolated moulds belonged to two different species: *Penicillium nalgiovense* (50 isolates), which predominated and was inoculated as a starter; *P. lanosoceruleum* (2 isolates) was also present (Table 6). Both species produced either TVB-N or acetic acid in vitro, but not biogenic amines. Either TVB-N or acetic acid contributed to the off-odour and off-flavour of the spoiled goose sausages, as perceived by the panellists.

The volatile compounds detected in spoiled and unspoiled sausages and their concentrations are displayed in Table 7 and are expressed as the ratio between the area of each peak and the area of the internal standard (4-methyl, 2-pentanol) from three analytical runs.

The volatile compounds detected in sausages were divided into aldehydes, ketones, spices (including garlic and pepper), alcohols, carboxylic acids, and other molecules.

Aldehydes were detected in low amounts and were slightly higher in the spoiled samples, mainly due to the concentration of hexanal, which results from lipid oxidation. Ketones accumulated in higher proportions in spoiled sausages: The molecules responsible for this difference were acetone, 2-butanone, and acetoin, whose concentrations were more than double those found in the unspoiled sausages. Among alcohols, in the spoiled samples, ethanol was detected at a concentration 10 times higher, but the amount of 1-propanol was extremely high compared to the control. Among acids, a similar behaviour characterised the presence of acetic acid. As expected, no relevant difference was observed among the molecules derived from spices, mainly terpenes derived from black pepper (limonene, caryophyllene, α- and β-pinene, 3-carene, α-phellandrene, and others) and garlic (allyl mercaptan, allyl methyl sulfide, diallyl disulfide, and other sulfur compounds).

However, the major ratio between the area of each peak and the area of the internal standard of ethanol and acetic acid observed in the spoiled with respect to the unspoiled sausages confirmed the activity of *Lev. brevis*, which is a heterofermentative LAB. Therefore, it could be hypothesised that either moulds or LAB could contribute to spoilage.

## 4. Discussion

Both the spoiled and unspoiled sausages were properly dried, as demonstrated by Aw; CPCNC and LAB concentrations were not significantly different (*p* > 0.05). On the other hand, the pH and the enterococci count differed significantly between spoiled and unspoiled sausages (*p* < 0.05). The level of enterococci values must be considered too high and is not regarded as normal for meat products [2,3,13,51]. In particular, the pH remained high despite the concentration of acidifying bacteria (LAB and Enterococci) and was similar to that usually found in sausages with defects [1,2,3,4].

The presence of a higher concentration of TVB-N in the spoiled goose sausages (302.5 mg N/100 g) than in the unspoiled sausages (*p* < 0.05) was demonstrated by visible and evident mould mycelium between the meat and casings. Iacumin et al. [1] showed a similar effect in spoiled goose sausages made in a facility in the Friuli region. The ammonia and acetic acid smell perceived either in the ripening area or in the spoiled sausages was confirmed by growing the isolated moulds in vitro, where both mould species were able to produce TVB-N and acetic acid. The high TVB-N concentration demonstrated spoilage because it is much lower in well-ripened Italian sausages. Usually, the TVB-N value is typically less than 100 mg N/100 g [17,52], as found in the unspoiled goose sausages, which contained a maximum level of 78.2 mg N/100 g. However, as demonstrated in previous works [1,17,53,54], it is also possible that enterococci, LAB, and CNCPC could have worked together with moulds in TVB-N production, considering that CPCNC and LAB can metabolise amino acids and produce TVB-N [55].

*Lev. brevis*, enterococci, and moulds prevailed during the ripening of spoiled goose sausages and produced further spoilage, determined by the presence of BA, volatile nitrogen (TVB-N), and acetic acid. In contrast, *Latilactobacillus sakei* and *Latilactobacillus curvatus*, representing the starter cultures added, were mainly isolated from unspoiled products, as expected. It could be hypothesised that in the spoiled sausages, the failure of the starter culture to develop favoured the growth and predominance of *Levilactobacillus brevis* and Enterococci, as shown by the values of enterococci in the spoiled sausages, which were 7 logs higher than those in the unspoiled ones. Conversely, the LAB concentration was quite similar at a level of 8.6–8.7 log CFU/g and did not show significant differences, but the strains of the spoiled samples were completely different from those of the unspoiled sausages.

Enterococci are known to be the most efficient tyramine producers [25,56,57]. In *E. faecalis*, the presence of tyrosine decarboxylase is a species characteristic, while it is widely diffused in *E. faecium* and in other species of the genus [1,58,59,60,61]. However, in this work, their ability to produce cadaverine and putrescine was demonstrated. The production of putrescine, derived from ornithine, can be linked to the arginine deiminase pathway, diffused among LAB [62,63]. The production of cadaverine by enterococci is less described in the literature.

In addition, many LAB show a strain-dependent ability to produce tyramine. In particular, *Levilactobacillus brevis* strains have been studied for their ability to produce this BA [64], but some strains can be responsible for the production of histamine, putrescine, and cadaverine [62,65,66]. Romano et al. [66] demonstrated the presence of a horizontally transferred acid resistance locus in some strains of this species, which may coexist with genes responsible for tyrosine and ornithine decarboxylases and malolactic enzymes. This possibility has been confirmed by the data reported in this work.

The BA production in the first instance depends on the presence of precursors (amino acids) and microorganisms possessing specific decarboxylases. Regarding the latter aspect, the concentration of microorganisms must reach rather high thresholds for the accumulation of BA in relevant amounts. From this perspective, fermented foods of animal origin are substrates characterised by a high potential risk [28]. The adoption of starter cultures without decarboxylase activity that can dominate the microbiota of sausages during fermentation and ripening is the most powerful tool to reduce this risk [67]. Nevertheless, when the initial microbial contamination of raw material is high, the presence of starter cultures cannot be sufficient to avoid the proliferation of undesirable microorganisms [68]. This was also confirmed in the present work, in which the species used as starter cultures were not isolated at the end of ripening. In addition, as shown by the LAB isolates from spoiled sausages, a predominance of decarboxylating microorganisms was demonstrated. The presence of Gram-negative bacteria (*Enterobacteriaceae*) in the ripened sausage seemed to be too low to justify an eventual role in the production of BA (especially cadaverine). Finally, it is well-recognised that some moulds can produce polyamines cadaverine and putrescine [28]. However, *Penicillium nalgiovense* and *P. lanosoceruleum*, the species isolated in the spoiled sausages, were not able to decarboxylate amino acids, as demonstrated by the in vitro tests.

The differences observed in the volatilome are the result of the metabolic activity of two distinct microbiota responsible for fermentation. In general, the overall sausage sensory profile is the result of several molecules. Odour defects often do not depend on the presence of a specific, extraneous molecule but rather on an imbalance in the relative ratios of the molecules present. Molecules such as 2-butanone are commonly present in the sausage volatilome; but at high concentrations, they may cause defects [69]. 2-Butanone may be produced through several bacterial pathways starting, for example, from pyruvate [70,71]. Pyruvate is also the precursor for molecules, such as acetone, diacetyl, and acetoin, which are always present in the volatile profile but in amounts that can be very different and have an impact on the sensory properties [72]. These metabolisms are often carried out by LAB and staphylococci, and obvious changes in their proportion reflect changes in the microbial population.

In the case studied here, all these compounds were found in higher amounts in spoiled sausages, indicating that the predominant species (Enterococci and *Lev. brevis*) were characterised by a more active metabolism in this perspective. The same consideration can be extended to the higher amount of acetic acid and ethanol found in spoiled samples. These are two typical end products of LAB mixed acid fermentation and heterolactic fermentation [73]. With these premises, the presence of *Lev. brevis* is sufficient for explaining the differences in the two concentrations of these molecules.

## 5. Conclusions

The investigated goose sausages were spoiled by a microorganism consortium that included *Levilactobacillus brevis*, enterococci, and moulds. The bacteria were responsible for high concentrations of some Bas, such as putrescine, tyrosine, and cadaverine, which caused spoilage. *Lev. brevis* was also responsible for ethanol and acetic acid production. The microbial consortium produced off-odours that were perceived through the “needle probing” technique and the off-flavour perceived by sniffing. Indeed, the high concentration of TVB-N and acetic acid, which resulted in the perception of ammonia and a light vinegar taste, has determined the defects of the spoilage sausages. Moulds were also recognised as responsible for spoilage. In particular, they grew between the meat and casing in the spoiled products, contributing either to the high ammonia or acetic acid smell, as demonstrated by in vitro tests.

Finally, it could be concluded that the control of the overall microbial groups during drying and ripening and the use of efficient starters will permit the production of safe goose sausages. Indeed, the starter cultures represented by *Latilactobacillus sakei* and *Latilactobacillus curvatus* grew and predominated only in the unspoiled products; inadequate growth was observed in the unspoiled sausages. Consequently, it could be hypothesised that in the spoiled sausages, the lack of development of the starter cultures favoured the growth and predominance of *Levilactobacillus brevis* and enterococci, leading to sausages spoilage.

## Figures and Tables

**Table 1 microorganisms-11-01942-t001:** The microbial concentration of spoiled and unspoiled sausages.

Microorganism	Unspoiled	Spoiled
Total aerobic count	7.8 ± 0.1 a	8.8 ± 0.2 b
Lactic acid bacteria	8.6 ± 0.2 a	8.7 ± 0.1 a
Yeasts	2.0 ± 0.1 a	2.2 ± 0.1 a
Moulds	2.3 ± 0.1 a	3.3 ± 0.2 b
Enterococci *	<100	7.5 ± 0.1 b
*Escherichia coli **	<10	<10
Enterobacteriaceae	2.1 ± 0.1 a	2.0 ± 0.2 a
CNCPC^1^	6.1 ± 0.2 a	5.8 ± 0.2 a
CCPPC^2^ *	<10	<10
Clostridia H_2_S+ *	<10	<10

CNCPC^1^, coagulase-negative, catalase-positive cocci; CCPPC^2^, coagulase-positive, catalase-positive cocci; data (CFU/g; * CFU/g) represent the means ± standard deviations of the total samples; means with the same letters within the same lane (following the values) are not significantly different (*p* < 0.05).

**Table 2 microorganisms-11-01942-t002:** Chemical–physical parameters of spoiled and unspoiled goose sausages.

Parameter	Unspoiled	Spoiled
pH	5.9 ± 0.2 a	6.3 ± 0.1 b
Aw	0.92 ± 0.01 a	0.92 ± 0.01 a
TVB-N ^	78.2 ± 9.5 a	302.5 ± 9.3 b
Lactic acid	120.4 ± 21.5 a	85.3 ±12.5 b
L*	38.2 ± 6.0 a	36.3 ± 4.1 a
a*	16.0 ± 1.2 a	17.6 ± 1.6 a
b*	1.2 ± 0.4 a	1.7 ± 0.9 a

TVB-N ^, total volatile basic nitrogen, mg N/100 g; lactic acid, mg/100 g. Data represent the means ± standard deviations of the total samples; means with the same letters within the same lane (following the values) are not significantly different (*p* < 0.05).

**Table 3 microorganisms-11-01942-t003:** Biogenic amines in spoiled and unspoiled goose sausages.

Biogenic Amines	Unspoiled	Spoiled
Histamine	5.6 ± 1.8 a	<LOD b
Putrescine	<LOD a	242.5 ± 39.9 b
Cadaverine	8.1 ± 1.2 a	510.4 ± 70.1 b
Spermine	<LOD	<LOD
Spermidine	<LOD	<LOD
Tyramine	<LOD a	388.3 ± 14.5 b
2-phenylethylamine	<LOD	<LOD

Biogenic amines, mg/kg; <L.O.D., limit of detection (3 mg/kg); data represent the means ± standard deviations of the total samples; means with the same letters within the same lane (following the values) are not significantly different (*p* < 0.05).

**Table 4 microorganisms-11-01942-t004:** Identification of LAB species isolated in spoiled (S) and unspoiled (U) sausages.

Agar	Strains	Source	Serial Decimal Dilution					
			10^−6^		10^−7^		10^−8^	
			U	S	U	S	U	S
MRS	*Latilactobacillus sakei*	*NR_113821.1*	58	20	42	5	30	-
	*Latilactobacillus curvatus*	*NR_113334.1*	4	6	-	1	-	-
	*Levilactobacillus brevis*	*AB362618.1*	-	52	-	30	-	15
KAA	*Enterococcus faecium*		-	20	-	6	-	-
	*Enterococcus faecalis*		-	10	-	3	-	-

MRS, De Man Rogosa Sharpe Agar; KAA, Kanamycin Aesculin Azide Agar; number of the isolated strains; - not found.

**Table 5 microorganisms-11-01942-t005:** Biogenic amine production in vitro.

Strains	Biogenic Amines			
	Tyrosine	Cadaverine	Putrescine	Histamine
*Latilactobacillus sakei*	*−*	−	−	−
*Latilactobacillus curvatus*	*+*	−	−	−
*Levilactobacillus brevis*	*+*	+	−	−
*Enterococcus faecium*	+	+	+	−
*Enterococcus faecalis*	+	+	+	−

+ positive production; − negative production.

**Table 6 microorganisms-11-01942-t006:** Identification of the strains isolated from the spoiled goose sausages and their production of TVB-N and acetic acid.

Identification	No. of Isolates	TBV-N/Acetic Acid/Biogenic Amines Production	Source
*Penicillium nalgiovense*	50	+/+/−	JQ434685.1
*Penicillium lanosocoeruleum*	2	+/+/−	NG069623.1

TVB-N, total volatile basic nitrogen; +, positive production; − negative production. The accession number of the closest related species was found by a BLAST search.

**Table 7 microorganisms-11-01942-t007:** Volatile compounds in unspoiled and spoiled goose sausages.

Compound	Unspoiled	Spoiled
Hexanal *	0.16 ± 0.02	0.51 ± 0.13
Nonanal	0.25 ± 0.03	0.23 ± 0.01
Benzaldehyde	0.10 ± 0.10	0.25 ± 0.18
Benzeneacetaldehyde	0.88 ± 0.15	0.90 ± 0.22
ALDEHYDES *	1.39 ± 0.16	1.89 ± 0.12
Acetone *	0.28 ± 0.13	1.05 ± 0.28
2-butanone *	1.56 ± 0.48	3.90 ± 3.07
2,3-butanedione	0.47 ± 0.03	0.45 ± 0.01
Methyl isobutyl ketone	2.75 ± 0.09	2.88 ± 0.06
2-hexanone, 4-methyl	0.40 ± 0.12	0.50 ± 0.05
4-heptanone *	0.26 ± 0.11	3.35 ± 0.21
3-hexen-2-one	2.90 ± 0.22	3.16 ± 1.58
Acetoin *	2.05 ± 0.01	4.87 ± 0.90
KETONES *	10.93 ± 0.66	20.16 ± 0.46
Isopropyl alcohol	0.25 ± 0.03	0.27 ± 0.01
Ethanol *	1.02 ± 0.21	11.33 ± 1.92
2-butanol *	1.85 ± 0.40	0.50 ± 0.01
1-propanol *	0.64 ± 0.03	2.13 ± 0.29
1-hexanol, 2-ethyl	0.09 ± 0.04	0.11 ± 0.01
2-hexanol	0.32 ± 0.15	0.30 ± 0.11
1-pentanol	0.46 ± 0.14	0.20 ± 0.04
1-hexanol	0.86 ± 0.11	0.98 ± 0.23
1-octen-3-ol	0.20 ± 0.09	0.23 ± 0.08
Heptanol	0.18 ± 0.04	0.16 ± 0.08
1-hexanol, 2-ethyl	0.09 ± 0.03	0.09 ± 0.01
1-octanol	0.11 ± 0.03	0.10 ± 0.01
Phenylethyl alcohol	0.34 ± 0.22	0.38 ± 0.18
ALCOHOLS *	6.41 ± 0.12	16.78 ± 0.66
Acetic acid *	0.95 ± 0.28	3.55 ± 1.00
Propanoic acid	0.11 ± 0.02	0.13 ± 0.02
ACIDS *	1.06 ± 0.26	3.68 ± 0.91
Styrene	1.55 ± 1.03	1.70 ± 1.80
Butyrolactone	0.33 ± 0.03	0.30 ± 0.07
OTHERS	1.88 ± 1.62	2.00 ± 0.65
Allyl mercaptan	1.09 ± 0.97	2.05 ± 1.70
Sulfide, allyl methyl	12.38 ± 1.02	15.58 ± 1.38
1-propene, 1-(methylthio)	0.55 ± 0.12	0.70 ± 0.07
α-pinene	0.50 ± 0.20	0.53 ± 0.30
disulfide, dimethyl	0.38 ± 0.22	0.42 ± 0.60
β-pinene	0.85 ± 0.07	0.83 ± 0.01
3-carene	1.93 ± 0.58	1.90 ± 0.90
α-phellandrene	2.04 ± 1.04	2.44 ± 1.54
Limonene	16.98 ± 3.18	18.72 ± 6.01
p-cymene	1.55 ± 0.11	1.64 ± 0.21
o-cymene	1.38 ± 0.23	1.25 ± 0.50
Terpinolene	0.34 ± 0.01	0.40 ± 0.11
4-carene	0.38 ± 0.10	0.31 ± 0.06
Diallyl disulphide	1.28 ± 0.11	1.22 ± 0.27
Copaene	0.90 ± 0.28	0.94 ± 0.66
Linalool	0.51 ± 0.12	0.56 ± 0.09
Caryophyllene	3.05 ± 0.40	3.13 ± 0.86
SPICES	46.19 ± 1.66	52.62 ± 3.23

Data (mean of 3 samples) expressed as the ratio between the area of each peak and the area of the internal standard (4-methyl, 2-pentanol); sum of compounds; data represent the means ± standard deviations (SD) of the total samples; compounds with a significantly different amount in relation to the two samples according to ANOVA (*p* ≤ 0.05) are marked with an asterisk.

## Data Availability

The data in this study are readily available upon reasonable request to the corresponding author.

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
