# Peer review of "Microbial Spoilage of Traditional Goose Sausages Produced in a Northern Region of Italy"

_microorganisms, 2023, doi:10.3390/microorganisms11081942_

Round 1

Reviewer 1 Report

The manuscript investigated the microorganisms and their metabolites responsible for the spoilage of goose sausages. It is an interesting study with some novelty and significance. However, I have some serious concerns about the research article and there are many points needed to be revised. The detailed comments are listed below.

1. Language is not acceptable. This paper should be corrected by a native speaker in English.

2. I suggest to revise the title of the manuscript. Generally speaking, title is not a sentence.

3. Line 9, please unify the format and size of the words.

4. Abstract, please add the words to explain why you carry out this research. What’s the significance of your study?

5. Line 51-53, please rewrite the sentence. There are some grammatical errors in this sentence.

6. The whole words in the “Introduction” should to be more concise and logical. Especially, the necessity and significance of this experiment should be clearly introduced in this part.

7. Line 99-117: Is the whole production process of spoiled goose sausage the same with normal goose sausage except the starter culture mix? If so, is there any possibility that the spoilage comes from the additional starter? Is it easier to analyze the difference between starter A and starter B for search the responsible microorganisms?

8. There were no figures in the whole manuscript. Are there any differences in appearances between the normal goose sausage and the spoiled ones? How did P. lanoso-coeruleum grow between the meat and casing? It’s best to exhibit the relative figures.

9. The presentation of the results was a bit confusing. I suggest to rewrite this part and exhibited the results more clearly.

10. The annotations of Table 1, Table 2 and Table 3 were a bit confusing. Please check and rewrite them.

11. There were 75 references in total. It is too many for a research article. Please remove the unimportant references. Moreover, please recheck and unify the format of all the references according to the author guidelines of Microorganisms.

12. Line 456-463, the sentences were talking about the security issues of the spoiled goose sausage. I suggest to transfer these sentences to the “Discussion” part.

13. Line 464-472, I suggest to rewrite this paragraph according the main conclusion of this research.

Language is not acceptable. This paper should be corrected by a native speaker in English.

Author Response

Dear reviewer

Enclosed you can find a copy of our revised Manuscript, enclosed you can find copy of the manuscript (Microorganisms - 2446039) entitled “Microbial spoilage of traditional goose sausages produced in a Northern Region of Italy”

Journal: Microorganisms

I add the answer to the referee.

The authors would like to thank the reviewers for their careful reading of the manuscript and the resulting constructive comments and suggestions. Basically, we agree with all of the points raised by the reviewers, and wherever possible the manuscript has been modified as recommended. All reviewer comments are in black plain font, whereas our response is described in red plain font.

We have made the changes and corrections on the basis of the reviewer’s suggestions. We evaluated the comments and prepared a point-by-point response to each one of them.

Reviewer 1

The manuscript investigated the microorganisms and their metabolites responsible for the spoilage of goose sausages. It is an interesting study with some novelty and significance. However, I have some serious concerns about the research article and there are many points needed to be revised. The detailed comments are listed below.

  1. Language is not acceptable. This paper should be corrected by a native speaker in English.

Answer – Thanks – The language paper has been corrected by AJE – as shown

  1. I suggest to revise the title of the manuscript. Generally speaking, title is not a sentence.

Answer – Thanks – I change the title – Microbial spoilage of traditional goose sausages produced in a Northern Region of Italy

  1. Line 9, please unify the format and size of the words.

Answer – Thanks – Lines 9 - I made it

  1. Abstract, please add the words to explain why you carry out this research. What’s the significance of your study?

Answer – Thanks I add what you suggest – Lines 14 – 17 - Recently, during the ripening of goose sausage, a defect consisting of ammonia and vinegar smell was noticed. The producer of the craft facility, located in Lombardia, a Northern region of Italy, asked us to identify the cause of that defect. Therefore, this study aimed to identify the potential responsible agents for the spoilage of this lot of goose sausages. of this lot of goose sausage produced in a small facility in Lombardy, a northern region of Italy.

  1. Line 51-53, please rewrite the sentence. There are some grammatical errors in this sentence.

Answer – Thanks – I changed the sentences – Lines 53 – 56- However, the choice of raw material, the natural microclimate of the drying/ripening rooms, and the aptitude of the producers allow to obtain a high quality product [1]. The goose sausage ripening depends on microbial and tissue enzyme activities and it is similar to that of sausages made with pork meat and fat [1-4].

  1. The whole words in the “Introduction” should to be more concise and logical. Especially, the necessity and significance of this experiment should be clearly introduced in this part.

Answer – Thanks – I made it  - Lines 33-39

In Italy, pork, beef, wild game (deer), and poultry meats are often used to produce a large number of traditional sausages. In Northern regions, from Lombardia to Friuli Venezia Giulia, goose meat represents one of the ­­­­­­­main ingredients for sausages, characterized by a sour taste and a semirigid consistency, which is elastic but not rubbery [1].

Goose products have always been present in the gastronomic tradition of different Northern Italy areas. The presence of Jewish communities in these areas has been documented since the fifteenth century and the most accredited hypothesis for the widespread presence of this type of sausage in these areas is the presence since the fifteenth century of Jewish communities whose religion forbids the consumption of pork meat; therefore, they had to consume meat coming from other animals like goose, whose breeding is favored by the characteristics of the territory.

Traditionally, the ingredients include fresh or frozen goose meat, NaCl, additives (nitrates, nitrites), spices, dextrose, and microbial starters, mainly represented by coagulase-negative catalase-positive cocci (CNCPC) and Latilactobacillus sakei. However, the recipe varies according to the producers. The quality of these sausages is not sufficiently standardized because they are mainly produced by shops, farms, typical restaurants, or by artisanal facilities, which are not equipped with appropriate drying and ripening chambers or systems with complete control of relative humidity (RH) and temperature. Therefore, each lot of production has its own history and can be completely different from other lots. However, the choice of raw material, the natural microclimate of the drying/ripening rooms, and the aptitude of the producers allow to obtain a high quality product [1]. The goose sausage ripening depends on microbial and tissue enzyme activities and it is similar to that of sausages made with pork meat and fat [1-4].

Among the microbial population, CNCPC and homofermentative lactic acid bacteria (LAB) are the main microorganisms responsible for ripening [1,5-10]. These bacteria originate from the raw meat, but they can be intentionally added as microbial starters to ensure consistent aroma and flavour, improve quality and safety, reduce the length of the curing period, and standardize the production [1-3,7,8,12-14]. Goose sausages are ancient and traditional products, and, recently, their demand has increased because consumers are searching for new foods; they represent an effort to generate alternatives to beef and pork meat products. While chicken sausages are often mixed with meat obtained from other animals, the combination of goose meat with other meat is quite rare; generally, goose sausages are produced using only goose meat [15]. In Italy, recently, the population has rediscovered products based on regional recipes, and goose sausages constitute an example [1]. The quality of craft sausages can depend on various parameters including goose breeding, slaughtering, and sausage manufacturing. However, goose breeding and slaughtering are easily checked depending on strict and established procedures. Indeed, in craft facilities, the main parameters influencing the quality are particularly caused by the production processes, which are often insufficiently checked, making the sausages unsuitable for consumption. The failure in the production process depends on the applied technology (Temperature/Relative Humidity of the ripening rooms), the natural microbiota of the meat, and the ingredients. The defects, coming from inadequate ripening, often lead to unpleasant odours or tastes. Ammonia, sulfur compounds, ketones, aldehydes, esters, organic acids, and biogenic amines represent the main molecules produced by spoilage microorganisms in sausages [1,16-19]. In particular, ammonia compounds lead to an increase in pH and to off-odours and off-flavours [1,17]. Enterobacteria, moulds, and LAB are often recognized as the responsible for producing ammonia and acetic acid, both indicators of sausage spoilage. Lactic acid bacteria can produce ammonia compounds only when a low level of sugar is added or present in the meat batter [16]; conversely, LAB produce lactic acid [1,16]. The presence of acetic acid, which gives sour smell to sausages, is produced by enterobacteria, moulds and heterofermentative LAB [1].

In fermented sausages, enterococci, Enterobacteriaceae and some LAB are usually the main producers of biogenic amines (BAs) such as tyramine, histamine, cadaverine, and putrescine [1,19-21], which, being vasoactive, at ingestion levels of 50–100 mg [22] can cause hypertension, migraine, brain haemorrhages, heart failure [23], urticaria, headache, flushing and abdominal cramps [20,24] in susceptible consumers.

Traditional microbiological methods, based on phenotypical aspects, are not useful for the early detection of BA producers in fermented sausages because only some strains of certain species can decarboxylate amino acids such as tyrosine or histidine [20,25]. Indeed, only by molecular methods it is possible to amplify all genes responsible for BA production [20,25-27]. Furthermore, in fermented sausages, BA formation depends on many factors including salt concentration, pH, quality of the raw materials, starter culture, technological additives, packaging, irradiation, high-pressure treatment, diameter of the sausages, an high initial concentration of decarboxylating microorganism, inadequate technological processes, and the possible presence of antimicrobial compounds in the spices mixture used in sausage production [1,19,25,29-35].

  1. Line 99-117: Is the whole production process of spoiled goose sausage the same with normal goose sausage except the starter culture mix? If so, is there any possibility that the spoilage comes from the additional starter? Is it easier to analyze the difference between starter A and starter B for search the responsible microorganisms?

Answer – Thanks – Lines 107-119 -  I wrote that the recipe and the technology applied were similar for both the lot (a and b), the only difference consisted of the lot of the starter  as shown: 

Two lots of sausages (50 each) with the same goose meat and recipe were produced by a small-scale facility located in Lombardy. These lots differed for the added starter culture mix (A and B), both composed of Staphylococcus xylosus and Latilactobacillus sakei (1/1 ratio) but of two different lots of production. The sausages had the following composition: goose meat, 97%; NaCl, 2.8%; KNO3, 0.02%; dextrose, 0.2%; black pepper, 0.002%; and nutmeg, 0.002%. Before adding the ingredient of the recipe, one lot was added with starter of lot A and one with starter of lot B at a final concentration of 6 log CFU/g. A starter culture of Penicillium nalgiovense was spread by aerosol (approximately 3 log CFU/cm2) onto casings. Natural casings derived from small intestine deprived of the mucosa, were used. The production process consisted of dehydration phase for 7 days with a Relative Humidity (RH) between 65 and 85% at 18-20 °C and of ripening phase for 40 days with a RH of approximately 70% at 15–17 °C.

  1. There were no figures in the whole manuscript. Are there any differences in appearances between the normal goose sausage and the spoiled ones? How did lanoso-coeruleum grow between the meat and casing? It’s best to exhibit the relative figures.

Answer – Thanks – I have not any figures but  the color of both the lots of products was similar as shown in table 2 about L*, a*, b* . No significative differences were observed. P. lanoso-coeruleum grew between the casing and the meat because in the spoiled sausages the dehydration left spaces between the meat and casing. In these space mould could grow. 

  1. The presentation of the results was a bit confusing. I suggest to rewrite this part and exhibited the results more clearly.

Answer – Thanks – We think that the presentation of the results is easily understandable and I do not find other words to improve the paper. According to my opinion the presentation of the results is not confusing. 

  1. The annotations of Table 1, Table 2 and Table 3 were a bit confusing. Please check and rewrite them.

Answer – Thanks – I modified it :

Lines 291 – 294 - Table 1 - CNCPC1: Coagulase Negative.Catalase Positive Cocci; CCPPC2: Coagulase Positive Catalase Positive Cocci;. Data (CFU/g; * CFU/g) represent the means ± standard deviations of the total samples; Mean with the same letters within the same lane (following the values) are not significantly differently (P< 0.05).

Table 2-3 the annotations are regular and used in all the papers, I examined.

  1. There were 75 references in total. It is too many for a research article. Please remove the unimportant references. Moreover, please recheck and unify the format of all the references according to the author guidelines of Microorganisms.

Answer – Thanks – I think that the references are adequate to better explain the works and in particular the discussion. I eliminated 74-75 references

  1. Line 456-463, the sentences were talking about the security issues of the spoiled goose sausage. I suggest to transfer these sentences to the “Discussion” part.

Answer – Thanks – I eliminate the security issue, which are not important for the aim of the paper.

  1. Line 464-472, I suggest to rewrite this paragraph according the main conclusion of this research.

Answer – Thanks – Lines 479-489 - The investigated goose sausages were spoiled by a microorganism consortium including Levilactobacillus brevis, Enterococci, and moulds. The bacteria were responsible for high concentrations of some BAs such as putrescine, tyrosine, and cadaverine, which caused spoilage. Lev. brevis was also responsible of ethanol and acetic acid production. The microbial consortium produced off-odour perceived through the “needle probing” technique and the off-flavour perceived by sniffing. Indeed, high concentration of TVB-N and acetic acid, which resulted in the perception of ammonia and a light vinegar taste determined the defects of the spoilage sausages. Moulds were also recognized as responsible for the spoilage. In particular, they grew between the meat and casing in the spoiled products contributing either to the high ammonia or acetic acid smell, as also demonstrated by the in vitro tests.

Reviewer 2 Report

the present study entitled "A strange spoilage in goose sausages and Levilactobacillus brevis, mould and enterococci are the responsible" is interesting and well organized. the authors have tried to give a meaningful conclusion. 

line 39-40: please add background and problem statement of the research, why this research is needed. 

line 94-96: objectives should be written in more details. 

line 103: write down the sausage making process not only composition. 

line 107: what natural casing? 

line 118: write more details about sensory analysis.

the paper is fine and can be considered for publication

Author Response

Dear reviewer

Enclosed you can find a copy of our revised Manuscript, enclosed you can find copy of the manuscript (Microorganisms - 2446039) entitled “Microbial spoilage of traditional goose sausages produced in a Northern Region of Italy”

Journal: Microorganisms

I add the answer to the referee.

The authors would like to thank the reviewers for their careful reading of the manuscript and the

resulting constructive comments and suggestions. Basically, we agree with all of the points raised

by the reviewers, and wherever possible the manuscript has been modified as recommended. All

reviewer comments are in black plain font, whereas our response is described in red plain font.

We have made the changes and corrections on the basis of the reviewer’s suggestions. We evaluated

the comments and prepared a point-by-point response to each one of them.

Reviewer 2

the present study entitled "A strange spoilage in goose sausages and Levilactobacillus brevis, mould and enterococci are the responsible" is interesting and well organized. the authors have tried to give a meaningful conclusion. 

line 39-40: please add background and problem statement of the research, why this research is needed. 

Answer – Thanks – Lines 38-44  - The investigated goose sausages were spoiled by a microorganism consortium including Levilactobacillus brevis, Enterococci, and moulds. The bacteria were responsible for high concentrations of some BAs such as putrescine, tyrosine, and cadaverine, which caused spoilage. Lev. brevis was also responsible of ethanol and acetic acid production. The microbial consortium produced off-odour perceived through the “needle probing” technique and the off-flavour perceived by sniffing. Indeed, high concentration of TVB-N and acetic acid, which resulted in the perception of ammonia and a light vinegar taste determined the defects of the spoilage sausages. Moulds were also recognized as responsible for the spoilage. In particular, they grew between the meat and casing in the spoiled products contributing either to the high ammonia or acetic acid smell, as also demonstrated by the in vitro tests.

I also modified the aim – Lines 100 - 104 - Recently, during the ripening of goose sausage, a defect consisting of ammonia and vinegar smell was noticed and subsequently confirmed by a sensory analysis made by untrained panelists. The producer of the craft facility asked us to identify the cause of the defect. Therefore, this study aimed to explore the potential responsible agents for the spoilage of this lot of goose sausage.

line 103: write down the sausage making process not only composition. 

Answer, Thanks – I add: Lines 117 - 119 - The production process consisted of dehydration phase for 7 days with a Relative Humidity (RH) between 65 and 85% at 18-20 °C and of ripening phase for 40 days with a RH of approximately 70% at 15–17 °C.

line 107: what natural casing? 

Answer – Thanks – Lines 115 – 116 - Natural casings derived from small intestine deprived of the mucosa, were used.

line 118: write more details about sensory analysis.

Answer – Thanks – I did not write the method because the difference between the two products was easily identified - I modified the sentence – Lines 136 – 144 - Then the presumed spoiled and unspoiled sausages were sliced and subject to sensory analysis using the triangle test methodology [ISO 4120:2004, Triangle test methodology. Standard test method for sensory analysis — General guidance for the design of test rooms]. Briefly: the untrained panelists were presented with three products, two of which were identical. Then they were asked to state which product they believed was a unique sample. The panelists who indicated by smelling the presence of two distinct samples were asked to identify which was the best or the acceptable sample, considering only the odor parameter.

Reviewer 3 Report

This manuscript investigated the microorganisms and their metabolites during the spoilage of goose sausages. My comments are as follows:

(1)    Line 51. “ripening (1,5-10].” It should be []

(2)    Line 67-68. “In sausages, spoilage microorganisms produce ammonia, sulfur compounds, ketones, aldehydes, esters and organic acids [1,16-18] and biogenic amines [16,19].” ” ketones, aldehydes, esters” These are produced in sausages for many reasons, and microorganisms are only one of the factors.

(3)    What are the conditions and parameters in the sausage ripening process?

(4)    Line 123. “…..carefully smelled by the panellists to identify the flavour and the odour and to try to determine the defect”. The description of Sensory analysis needs to draw on professional articles and specifications, and this part of the presentation needs to be further enhanced.

(5)    Line 167. “(gas pack anaerobic system, BBL; Becton Dickinson, USA)” Inconsistent font size

(6)    In Table 1. Enterobacteriaceae, why Unspoiled is less than Spoiled?

(7)    In Table 2. Why is there no significant difference between Unspoiled and Spoiled in Aw?

(8)    In Table 7. Why are the data without units and significance analysis?

(9)    Line 359. “mainly due to the concentration of hexanal, which results from lipid oxidation.” Is the degree of lipid oxidation different between these two groups?

(10) The process of sausage spoilage is very complicated, do you think your important findings can make this process clear to the reader?

(11) The number of references is quite a lot, but the last 3-5 years are too few

Author Response

Dear reviewer

Enclosed you can find a copy of our revised Manuscript, enclosed you can find copy of the manuscript (Microorganisms - 2446039) entitled “Microbial spoilage of traditional goose sausages produced in a Northern Region of Italy”

Journal: Microorganisms

I add the answer to the referee.

The authors would like to thank the reviewers for their careful reading of the manuscript and the

resulting constructive comments and suggestions. Basically, we agree with all of the points raised

by the reviewers, and wherever possible the manuscript has been modified as recommended. All

reviewer comments are in black plain font, whereas our response is described in red plain font.

We have made the changes and corrections on the basis of the reviewer’s suggestions. We evaluated

the comments and prepared a point-by-point response to each one of them.

Reviewer 3

  • Line 51. “ripening (1,5-10].” It should be []

Answer – Thanks I made it – Line 58

  • Line 67-68. “In sausages, spoilage microorganisms produce ammonia, sulfur compounds, ketones, aldehydes, esters and organic acids [1,16-18] and biogenic amines [16,19].” ” ketones, aldehydes, esters” These are produced in sausages for many reasons, and microorganisms are only one of the factors.

Answer – Thanks – I change the sentence – lines 76-78 - Ammonia, sulfur compounds, ketones, aldehydes, esters, organic acids, and biogenic amines represent the main molecules produced by spoilage microorganisms in sausages [1,16-19].

  • What are the conditions and parameters in the sausage ripening process?

Anwer – Thanks – Lines 117-119 - The production process consisted of dehydration phase for 7 days with a Relative Humidity (RH) between 65 and 85% at 18-20 °C and of ripening phase for 40 days with a RH of approximately 70% at 15–17 °C.

  • Line 123. “…..carefully smelled by the panellists to identify the flavour and the odour and to try to determine the defect”. The description of Sensory analysis needs to draw on professional articles and specifications, and this part of the presentation needs to be further enhanced.

Answer – Thanks – I did not write the method because the difference between the two products was easily identified - I modified the sentence – Lines 136 – 143 -  Then the presumed spoiled and unspoiled sausages were sliced and subject to sensory analysis using the triangle test methodology [ISO 4120:2004, Triangle test methodology. Standard test method for sensory analysis — General guidance for the design of test rooms]. Briefly: the untrained panelists were presented with three products, two of which were identical. Then they were asked to state which product they believed was a unique sample. The panelists who indicated by smelling the presence of two distinct samples were asked to identify which was the best or the acceptable sample, considering only the odor parameter.

  • Line 167. “(gas pack anaerobic system, BBL; Becton Dickinson, USA)” Inconsistent font size

Answer – Thanks – Line 187 - I specified – (BD BBL 261205 CO2 generators – GasPack jars - Becton Dickinson, USA).

  • In Table 1. Enterobacteriaceae, why Unspoiled is less than Spoiled?

Answer – Thanks. I made a mistake –Line 290 -  Both the loads are similar - no statistical differences exist (p > 0.05). I correct it in table 1. Line 275 - Enterobacteriaceae did not (p > 0.05).

  • In Table 2. Why is there no significant difference between Unspoiled and Spoiled in Aw?

Answer – Thanks – The ripening methods were similar, consequently the value of Aw at the end or ripening were similar. Lines 303 - 304 - Aw was similar at a level of 0.92, depending on the ripening methods, which was similar for the spoiled and unspoiled sausages.

In Table 7. Why are the data without units and significance analysis?

Answer – Thanks – Lines 373 - 376 - Data (mean of 3 samples) expressed as the ratio between the area of each peak and the area of the internal standard (4-methyl, 2-pentanol); Sum of compounds; Data represent the means ± standard deviations (SD) of the total samples; Compounds with significantly different amount in relation to the two samples according to ANOVA (p ≤ 0.05) are marked with an asterisk.

  • Line 359. “mainly due to the concentration of hexanal, which results from lipid oxidation.” Is the degree of lipid oxidation different between these two groups?

Answer – Thanks – the degree of lipid oxidation was not determined for both the products, because the work was focused on the spoilage caused by microorganisms. So, we hypothesized that the different hexanal concentration could be due on lipid oxidation.

  • The process of sausage spoilage is very complicated, do you think your important findings can make this process clear to the reader?

Answer – Thanks – I think that our findings can give a little help in the understanding of a particular type of spoilage in the goose sausages.

Reviewer 4 Report

The current manuscript reports a strange spoilage in goose sausages and Levilactobacillus brevis, mould and enterococci are the responsible.

In general, this is an important and interesting research, logically structured. The research methods used are described in detail, the results are discussed.

I have a few comments and suggestions.

In the section «Goose sausage products analyzed», briefly describe the manufacturing technology of the product, indicating the technological parameters.

In the section «Statistical analysis», you must specify the number of repetitions.

In the Discussion, the authors duplicate the description of the results presented in the tables.

References should be removed from the Conclusion. In the Conclusion, add what your results mean and who they are for.

Author Response

Dear reviewer

Enclosed you can find a copy of our revised Manuscript, enclosed you can find copy of the manuscript (Microorganisms - 2446039) entitled “Microbial spoilage of traditional goose sausages produced in a Northern Region of Italy”

Journal: Microorganisms

I add the answer to the referee.

The authors would like to thank the reviewers for their careful reading of the manuscript and the

resulting constructive comments and suggestions. Basically, we agree with all of the points raised

by the reviewers, and wherever possible the manuscript has been modified as recommended. All

reviewer comments are in black plain font, whereas our response is described in red plain font.

We have made the changes and corrections on the basis of the reviewer’s suggestions. We evaluated

the comments and prepared a point-by-point response to each one of them.

Reviewer 4

The current manuscript reports a strange spoilage in goose sausages and Levilactobacillus brevis, mould and enterococci are the responsible.

In general, this is an important and interesting research, logically structured. The research methods used are described in detail, the results are discussed.

I have a few comments and suggestions.

In the section «Goose sausage products analyzed», briefly describe the manufacturing technology of the product, indicating the technological parameters.

Answer – Thanks – Lines 117 – 119 - I add the technological parameters - The production process consisted of dehydration phase for 7 days with a Relative Humidity (RH) between 65 and 85% at 18-20 °C and of ripening phase for 40 days with a RH of approximately 70% at 15–17 °C.

In the section «Statistical analysis», you must specify the number of repetitions.

Answer – Thanks – I add Lines 237-238 - Each physical-chemical and microbial analysis included ten samples either for spoiled and unspoiled goose sausages. Three samples were tested for volatilome analysis.

In the Discussion, the authors duplicate the description of the results presented in the tables.

Answer – Thanks – I know but it is necessary for the discussion

References should be removed from the Conclusion. In the Conclusion, add what your results mean and who they are for.

Answer – Thanks – I eliminated the references.

Reviewer 5 Report

The title of the manuscript should be rephrased -it is not clear and a little confused

L 36-39 in lines afore you mentioned that the recipe varies, and after that you included all ingredients by % that are usually (traditionally) used for this sausage manufacturing? This is also contradictory to the lines 41-45

L63-91..not very well introduced...you mentioned numerous quality problems related to sausage quality and after that described (focused) only on spoilage microorganisms and their byproducts...this part should be rephrased

M&m section - must be rephrased...please describe in detail goose sausage production..describe differences in those two lots..why these receptures were choosen..how did you finally get spoiled and unspoiled lots?

L 108-109...not clear, how did you get these two lots,  spoiled and unspoiled?

L110-111...not clear-you described final step and after that the ripenning phase?

2.2. Sensory analysis must be desribed in detail...e.g. room condition, sample preparation, used methodology, replication number etc.

L 126-133..must be described detailed 

L235-238...please desribe in m&m section in detail or supplement with additional material

Discussion - there is no proper discussion in the manuscript..just co.parison wizh another studies

Conclusion..is not clear..please do not repeat results from the study...clearly conclude from your results!

Author Response

Dear reviewer

Enclosed you can find a copy of our revised Manuscript, enclosed you can find copy of the manuscript (Microorganisms - 2446039) entitled “Microbial spoilage of traditional goose sausages produced in a Northern Region of Italy”

Journal: Microorganisms

I add the answer to the referee.

The authors would like to thank the reviewers for their careful reading of the manuscript and the

resulting constructive comments and suggestions. Basically, we agree with all of the points raised

by the reviewers, and wherever possible the manuscript has been modified as recommended. All

reviewer comments are in black plain font, whereas our response is described in red plain font.

We have made the changes and corrections on the basis of the reviewer’s suggestions. We evaluated

the comments and prepared a point-by-point response to each one of them.

Reviewer 5

The title of the manuscript should be rephrased -it is not clear and a little confused.

Answer – Thanks – I change the title – Microbial spoilage of traditional goose sausages produced in a Northern Region of Italy

L 36-39 in lines afore you mentioned that the recipe varies, and after that you included all ingredients by % that are usually (traditionally) used for this sausage manufacturing? This is also contradictory to the lines 41-45.

Answer – Thanks – The ingredients used are quite similar but the concentrations are variable. So I modified eliminating the concentrations of the ingredients – Lines 41 - 48 - Traditionally, the ingredients include fresh or frozen goose meat, NaCl, additives (nitrates, nitrites), spices, dextrose, and microbial starters, mainly represented by coagulase-negative catalase-positive cocci (CNCPC) and Latilactobacillus sakei. However, the recipe varies according to the producers

L63-91..not very well introduced...you mentioned numerous quality problems related to sausage quality and after that described (focused) only on spoilage microorganisms and their byproducts...this part should be rephrased

Answer – Thanks – Lines 68-78 - The quality of craft sausages can depend on various parameters including goose breeding, slaughtering, and sausage manufacturing. However, goose breeding and slaughtering are easily checked depending on strict and established procedures. Indeed, in craft facilities, the main parameters influencing the quality are particularly caused by the production processes, which are often insufficiently checked, making the sausages unsuitable for consumption. The failure in the production process depends on the applied technology (Temperature/Relative Humidity of the ripening rooms), the natural microbiota of the meat, and the ingredients. The defects, coming from inadequate ripening, often lead to unpleasant odours or tastes. Ammonia, sulfur compounds, ketones, aldehydes, esters, organic acids, and biogenic amines represent the main molecules produced by spoilage microorganisms in sausages [1,16-19]

M&m section - must be rephrased...please describe in detail goose sausage production..describe differences in those two lots..why these receptures were choosen..how did you finally get spoiled and unspoiled lots?

Answer – Thanks – Lines 107 – 119 - In the material e methods – It was reported that both the lots were made with the same ingredients, the same technology (Which was added). The recipe is similar and typical of that craft facility. They differed only for the lot of the starter. The starter has the same microbial composition, was used at the same concentration. 

Two lots of sausages (50 each) with the same goose meat and recipe were produced by a small-scale facility located in Lombardy. These lots differed for the added starter culture mix (A and B), both composed of Staphylococcus xylosus and Latilactobacillus sakei (1/1 ratio) but of two different lots of production. The sausages had the following composition: goose meat, 97%; NaCl, 2.8%; KNO3, 0.02%; dextrose, 0.2%; black pepper, 0.002%; and nutmeg, 0.002%. Before adding the ingredient of the recipe, one lot was added with starter of lot A and one with starter of lot B at a final concentration of 6 log CFU/g. A starter culture of Penicillium nalgiovense was spread by aerosol (approximately 3 log CFU/cm2) onto casings. Natural casings derived from small intestine deprived of the mucosa, were used. The production process consisted of dehydration phase for 7 days with a Relative Humidity (RH) between 65 and 85% at 18-20 °C and of ripening phase for 40 days with a RH of approximately 70% at 15–17 °C.

L 108-109...not clear, how did you get these two lots,  spoiled and unspoiled?

Answer – Thanks – Lines 123 – 127 - During the ripening, the head of the sausage production checked both lots, but ten days before the end of ripening, which lasted 50 days, the sausages with added lot B starter gave off an ammonia and vinegar smell, which was also widespread in the ripening rooms. Consequently, he identified this lot b as spoiled and asks us to identify the causes. The defect was confirmed by the sensory analysis listed below.

L110-111...not clear-you described final step and after that the ripening phase?

Answer – Thanks – Lines 117 - 119 - The production process consisted of dehydration phase for 7 days with a Relative Humidity (RH) between 65 and 85% at 18-20 °C and of ripening phase for 40 days with a RH of approximately 70% at 15–17 °C

2.2. Sensory analysis must be desribed in detail...e.g. room condition, sample preparation, used methodology, replication number etc.

Answer – Thanks – Lines 136 - 143 - Then the presumed spoiled and unspoiled sausages were sliced and subject to sensory analysis using the triangle test methodology [ISO 4120:2004, Triangle test methodology. Standard test method for sensory analysis — General guidance for the design of test rooms]. Briefly: the untrained panelists were presented with three products, two of which were identical. Then they were asked to state which product they believed was a unique sample. The panelists who indicated by smelling the presence of two distinct samples were asked to identify which was the best or the acceptable sample, considering only the odor parameter.

L 126-133..must be described detailed 

Answer – Thanks – the Pearson method was not modified, consequently I wrote briefly the method as just made in  other works – Lines 146 – 148  - Briefly, TVB-N was released by boiling the sample directly with magnesium oxide, which also prevented volatile acids from distilling over into boric acid. The distillate was titrated with standard acid. 

L235-238...please desribe in m&m section in detail or supplement with additional material

Answer – Thanks – I do not have supplement material, Lines 250 - 255 - Data were analysed using Statistica 7.0 vers. 8 software (Statsoft Inc., Tulsa, OK, USA, 2008). The values of the different parameters were compared by one-way analysis of variance, and the means were then compared using Tukey’s honest significance test. Differences were considered significant at p < 0.05. Each physical-chemical and microbial analysis included ten samples either for spoiled and unspoiled goose sausages. Three samples were tested for volatilome analysis.

Discussion - there is no proper discussion in the manuscript..just co.parison wizh another studies

Answer – Thanks – According my opinion the discussion is adequate, and speaks about the obtained data, giving an explanation of the causes of the defect. In addition, I compare the results with other studies.  The used references appear adequate and permit to explain the results and the discussion.

Conclusion..is not clear..please do not repeat results from the study...clearly conclude from your results!

Answer – Thanks – I modified the conclusion – Lines 480 – 490

The investigated goose sausages were spoiled by a microorganism consortium including Levilactobacillus brevis, Enterococci, and moulds. The bacteria were responsible for high concentrations of some BAs such as putrescine, tyrosine, and cadaverine, which caused spoilage. Lev. brevis was also responsible of ethanol and acetic acid production. The microbial consortium produced off-odour perceived through the “needle probing” technique and the off-flavour perceived by sniffing. Indeed, high concentration of TVB-N and acetic acid, which resulted in the perception of ammonia and a light vinegar taste determined the defects of the spoilage sausages. Moulds were also recognized as responsible for the spoilage. In particular, they grew between the meat and casing in the spoiled products contributing either to the high ammonia or acetic acid smell, as also demonstrated by the in vitro tests.

Round 2

Reviewer 1 Report

The revised manuscript is still not fit to be published in "Microorganism". I suggest the authors to add more data to support the conclusion.

Author Response

Dear reviewer

Enclosed you can find a copy of our revised Manuscript, enclosed you can find copy of the manuscript (Microorganisms - 2446039) entitled “Microbial spoilage of traditional goose sausages produced in a Northern Region of Italy”

Journal: Microorganisms

I add the answer to the referee.

The authors would like to thank the reviewers for their careful reading of the manuscript and the resulting constructive comments and suggestions. Basically, we agree with all of the points raised by the reviewers, and wherever possible the manuscript has been modified as recommended. All reviewer comments are in black plain font, whereas our response is described in red plain font.

We have made the changes and corrections on the basis of the reviewer’s suggestions. We evaluated the comments and prepared a point-by-point response to each one of them.

Reviewer 1:

The revised manuscript is still not fit to be published in "Microorganism". I suggest the authors to add more data to support the conclusion.

Answer, Thanks – Lines 480 – 498 - The investigated goose sausages were spoiled by a microorganism consortium that included Levilactobacillus brevis, enterococci, and moulds. The bacteria were responsible for high concentrations of some BAs such as putrescine, tyrosine, and cadaverine, which caused spoilage. Lev. brevis was also responsible for ethanol and acetic acid production. The microbial consortium produced off-odours that were perceived through the “needle probing” technique and the off-flavour perceived by sniffing. Indeed, a high concentration of TVB-N and acetic acid, which resulted in the perception of ammonia and a light vinegar taste have determined the defects of the spoilage sausages.

Moulds were also recognized as responsible for spoilage. In particular, they grew between the meat and casing in the spoiled products contributing either to the high ammonia or acetic acid smell, as demonstrated by in vitro tests.

Finally, it could be concluded that the control of the overall microbial groups during drying and ripening and the use of efficient starter will permit the production of safe goose sausages. Indeed, the starter cultures added represented by Latilactobacillus sakei and Latilactobacillus curvatus, grew and predominated only in the unspoiled products, inadequate growth was observed in the unspoiled sausages. Consequently, it could be hypothesized that in the spoiled sausages, the lack of development of the starter cultures favoured the growth and predominance of Levilactobacillus brevis and enterococci, leading to sausages spoilage.

Reviewer 2 Report

no further comments

fine

Author Response

Dear reviewer

Enclosed you can find a copy of our revised Manuscript, enclosed you can find copy of the manuscript (Microorganisms - 2446039) entitled “Microbial spoilage of traditional goose sausages produced in a Northern Region of Italy”

Journal: Microorganisms

I add the answer to the referee.

The authors would like to thank the reviewers for their careful reading of the manuscript and the resulting constructive comments and suggestions. Basically, we agree with all of the points raised by the reviewers, and wherever possible the manuscript has been modified as recommended. All reviewer comments are in black plain font, whereas our response is described in red plain font.

We have made the changes and corrections on the basis of the reviewer’s suggestions. We evaluated the comments and prepared a point-by-point response to each one of them.

Reviewer 2:

no further comments

Answer – Thanks

Reviewer 3 Report

The quality of the manuscript is greatly improved to the point where it can be considered for acceptance. One small question or suggestion.

Line 159-162. L*, a*, and b*. The full name can be written on the first occurrence and abbreviated on subsequent occurrences

Author Response

Dear reviewer

Enclosed you can find a copy of our revised Manuscript, enclosed you can find copy of the manuscript (Microorganisms - 2446039) entitled “Microbial spoilage of traditional goose sausages produced in a Northern Region of Italy”

Journal: Microorganisms

I add the answer to the referee.

The authors would like to thank the reviewers for their careful reading of the manuscript and the resulting constructive comments and suggestions. Basically, we agree with all of the points raised by the reviewers, and wherever possible the manuscript has been modified as recommended. All reviewer comments are in black plain font, whereas our response is described in red plain font.

We have made the changes and corrections on the basis of the reviewer’s suggestions. We evaluated the comments and prepared a point-by-point response to each one of them.

Reviewer 3

Line 159-162. L*, a*, and b*. The full name can be written on the first occurrence and abbreviated on subsequent occurrences

Answer – Thanks – Lines 158-159 - The evaluated parameters were Lightness (L*), red/green chromaticity (a*), and yellow/blue chromaticity (b*).

Reviewer 5 Report

No further comments.

Author Response

Dear reviewer

Enclosed you can find a copy of our revised Manuscript, enclosed you can find copy of the manuscript (Microorganisms - 2446039) entitled “Microbial spoilage of traditional goose sausages produced in a Northern Region of Italy”

Journal: Microorganisms

I add the answer to the referee.

The authors would like to thank the reviewers for their careful reading of the manuscript and the resulting constructive comments and suggestions. Basically, we agree with all of the points raised by the reviewers, and wherever possible the manuscript has been modified as recommended. All reviewer comments are in black plain font, whereas our response is described in red plain font.

We have made the changes and corrections on the basis of the reviewer’s suggestions. We evaluated the comments and prepared a point-by-point response to each one of them.

Reviewer 5

no further comments

Answer - Thanks
